# DNA barcoding of brackish and marine water fishes and shellfishes of Sundarbans, the world's largest mangrove ecosystem

Kazi Ahsan Habib[1,2]⊕*, Amit Kumer Neogi[2]⊕, Muntasir Rahman[3], Jina Oh[4], Youn-Ho Lee[4], Choong-Gon Kim[4]

1 Faculty of Fisheries, Department of Fisheries Biology and Genetics, Aquaculture and Marine Science, Sher-e-Bangla Agricultural University, Dhaka, Bangladesh, 2 Department of Fisheries Biology and Genetics, Aquatic Bioresource Research Lab, Sher-e-Bangla Agricultural University, Dhaka, Bangladesh, 3 Department of Biological Science, Wayne State University, Detroit, Michigan, United States of America, 4 Marine Biology and Biological Oceanography Division, Korea Institute of Ocean Science and Technology (KIOST), Busan, Korea

⊕ These authors contributed equally to this work.
* ahsan.sau@gmail.com, habibka@sau.edu.bd

**Data Availability Statement:** DNA sequences, Trace file, are publicly accessible on the Barcode of Life Data Systems (http://v3.boldsystems.org/) under the projects name: SUN, SAUR, CRU, MSK following the protocol outlined in the Methods

## Abstract

The present study aims to apply a DNA barcoding tool through amplifying two mitochondrial candidate genes i.e., COI and 16S rRNA for accurate identification of fish, aquatic molluscs and crustaceans of Sundarbans mangrove wetland, to build a reference library of fish and shellfishes of this unique ecosystems. A total of 185 mitochondrial COI barcode sequences and 59 partial sequences of the 16S rRNA gene were obtained from 120 genera, 65 families and 21 orders of fish, crustaceans and molluscs. The collected samples were first identified by examining morphometric characteristics and then assessed by DNA barcoding. The COI and 16S rRNA sequences of fishes and crustaceans were clearly discriminated among genera in their phylogenies. The average Kimura two-parameter (K2P) distances of COI barcode sequences within species, genera, and families of fishes are 1.57±0.06%, 15.16±0.23%, and 17.79±0.02%, respectively, and for 16S rRNA sequences, these values are 1.74±.8%, 0.97±.8%, and 4.29±1.3%, respectively. The minimum and maximum K2P distance based divergences in COI sequences of fishes are 0.19% and 36.27%, respectively. In crustaceans, the K2P distances within genera, families, and orders are 1.4±0.03%, 17.73±0.15%, and 22.81±0.02%, respectively and the minimum and maximum divergences are 0.2% and 33.93%, respectively. Additionally, the present study resolves the misidentification of the mud crab species of the Sundarbans as *Scylla olivacea* which was previously stated as *Scylla serrata*. In case of molluscs, values of interspecific divergence ranges from 17.43% to 66.3% in the barcoded species. The present study describes the development of a molecular and morphometric cross-referenced inventory of fish and shellfish of the Sundarbans. This inventory will be useful in future biodiversity studies and in forming future conservation plan.

section. In addition, all barcode sequences were deposited on NCBI GenBank under the separate accession no. given in S1 Table.

**Funding:** The author(s) received funding for this work from the Yeosu project funded by Expo 2012 Yeosu Korea Foundation in 2017. The funders had no role in study design, data collection and analysis, decision to publish, or preparation of the manuscript.

**Competing interests:** The authors have declared that no competing interests exist.

## Introduction

Mangroves are among some of the most productive and biologically diverse ecosystems on the planet, and create unique ecological environments for variety of plants, birds, reptiles, mammals and aquatic fauna. These ecospheres provide vital and unique ecosystem goods and services to human society, and coastal and marine systems [1]. Mangroves are mainly found along tropical and sub-tropical coastal regions of the world [2]. These intertidal wetland ecosystems support a complex aquatic food web [3]. The waters surrounding mangroves are a rich source of fish and shellfish. Mangroves enhance fishery production through two key mechanisms: the provision of food and of shelter Decomposition of their leaves and woody matter (detritus) form a major part of the coastal and marine food chains that supports fisheries [4,5]. The aquatic faunas of mangroves are uniquely adapted to survive under wide ranges of salinities. Besides, they are also well adapted with tidal amplitudes, temperatures and silty turbid water and hypoxic waterlogged conditions. The rich fish diversity in mangrove systems derives from diversified habitats such as mudflats and different adjoining water bodies such as rivers, estuaries, bays, intertidal creeks, channels and backwaters [6]. Mangrove areas also act as important breeding and nursery habitat for many species of fishes and invertebrates such as oysters, shrimp, and crab due to high density of organic matter deposition in a relatively small area [7].

The Sundarbans, the world's single largest continuous mangrove forest, lies in the southwest of Bangladesh and the southeastern portion of the state of West Bengal in India with a total area of about 10,000 sq. km [8,9]. In Bangladesh, the Sundarbans has an area of some 6,017 $km^2$ (7,620 $km^2$ including the marine zone) [9]. The Sundarbans, Bangladesh was declared as Ramsar Site (i.e. wetland of international importance) in 1992 and inscribed as Natural World Heritage Site (WHS) by UNESCO in 1997. The water area of the Sundarbans is about 2,000 $km^2$, which is about 33% of the total forest [10]. The impact of different anthropogenic activities like destructive fishing practices (using harmful fishing gears and poisoning), water pollution (industrial, oil spillage) and ever-increasing human population affect the aquatic ecosystem of Sundarbans. All of these factors highlight the urgent need for comprehensive assessments of aquatic biodiversity of Sundarbans to provide information on species diversity that will help in management and conservation of the fish and shellfish fauna [10–12]. Due to the existence of numerous cross-connecting channels linking catchment areas and estuaries, the pattern of river salinity in the Sundarbans is very intricate. Thus, this biome provides a home for several numbers of different aquatic species including fish, shrimp, crabs and molluscs [10,11]. Many of these species are economically important, many exhibits migratory behaviors, and some are at risk of extinction. Considering the importance of aquatic fauna of the Sundarbans and for creating their effective management and conservation strategy, the present study aims to assemble a comprehensive reference DNA barcode library of brackish and marine water fish and shellfish species. This kind of study by DNA barcoding within mangroves ecosystem is a rarity in south and Southeast Asia including Bangladesh.

DNA sequence analysis has been used to assist species identifications for different taxonomic groups and recently gained attention under the terms DNA barcoding or DNA taxonomy [13–15]. DNA barcoding refers to rapid, accurate species identification using one or a few short, standardized DNA regions [16–18]. These sequences tend to vary between species but are relatively invariable within species, a reference sequence library derived from correctly identified reference organisms can be used to identify unknown specimens and by matching with library [19]. DNA barcode information are also essential to resolve cryptic species complexes and for identifying conservation units within species [20,21]. Thus, this method now

represents the largest effort to catalogue biodiversity using molecular approaches. Hebert et al. [22] proposed that mitochondrial cytochrome c oxidase I (COI) gene can serve as a sufficient molecular tool to differentiate all, or at least the vast majority of, animal species. Afterward, the 5′ region of the COI mitochondrial gene has been used as a universal marker for large-scale identification of organisms in ecological or genomic studies [23–28]. Mitochondrial 16S ribosomal RNA (16S rRNA) gene region also has been used for identification of different organisms including marine invertebrates and fishes [11,29–35]. 16S rRNA gene is highly conserved among the animal taxa [29]. Although the absolute rate of change in the 16S rRNA gene sequence is not known, it does mark evolutionary distance and relatedness of organisms [36–39]. Alternatively, the choice between mitochondrial COI and 16S rRNA gene regions as a standard barcode is sometimes an issue of debate for some taxa [32]. However, we explore the 16S rRNA gene as an additional tool with the COI gene in DNA barcoding of fishes and shellfishes (crustaceans and molluscs) of the Sundarbans.

## Materials and methods

### Specimen collection and sampling area

The specimens of fishes, crustaceans and molluscs were sampled from the tidal rivers, canals and coasts at different locations of Sundarbans between July 2015 and June 2017. Samples were collected from local fishermen groups covering the uses of different fishing gears such as gill nets, lift nets, slat traps, hoop nets, angling. Samples were also collected from the fish landing sites inside or adjacent to the Sundarbans of Khulna, Bagherhat and Satkhira districts. Mollusc samples were collected from intertidal mudflat of the channels and rivers. The fish and crustacean (shrimps and crabs) samples of marine waters of the Bay of Bengal adjacent to the Sundarbans were collected from fishing village at Alorkol of Dubla Island located in the south coast of the Sundarbans, Bangladesh during winter season from November to March of each study year. This field research granted permission from Bangladesh Forest Department, Ministry of Forest, Environment and Climate change, Bangladesh (Memo no: 22.01.0000.004.04.21.2.15.458; dated: 11.05.2015)". Animals were handled with maximum care to minimize injuries during studies. The authors followed all applicable national and/or institutional guidelines for testing, care, and the use of animals. At least three individuals were tried to collect for each species. However, single individual was also collected for rare species. Upon collection and morphological sorting, specimens were identified to the extent possible in the field. The locations involved in the study were not part of any protected area or channels of the Sundarbans.

Specimens were sampled from a wide area of the Sundarbans (Fig 1). Collected samples were transferred using icebox with crushed ice to the "Field Laboratory and Research Station" temporarily established for the project implementation at Khulna, Bangladesh. Specimens were photographed and labelled in the lab and stored in the refrigerator at –20˚C. A small piece (5–7 mm$^3$) of muscle or fin tissue for fishes was removed from each fresh specimen and preserved in a sterile 1.5ml tube containing 96% ethanol at –20˚C for subsequent molecular work. Occasionally, tissues were collected directly from the fish landing sites for big-sized and endangered fishes with taking photographs. After such primary processing, all the frozen specimens and tissue samples were carried to the Aquatic Bioresource Research laboratory (ABR Lab.) in the Department of Fisheries Biology and Genetics of Sher-e-Bangla Agricultural University (SAU), Dhaka for further morphometric and molecular analyses. The voucher specimens were fixed with 95% ethanol and stored in the Marine Sample Museum at the Faculty of Fisheries, Aquaculture and Marine Science, SAU.

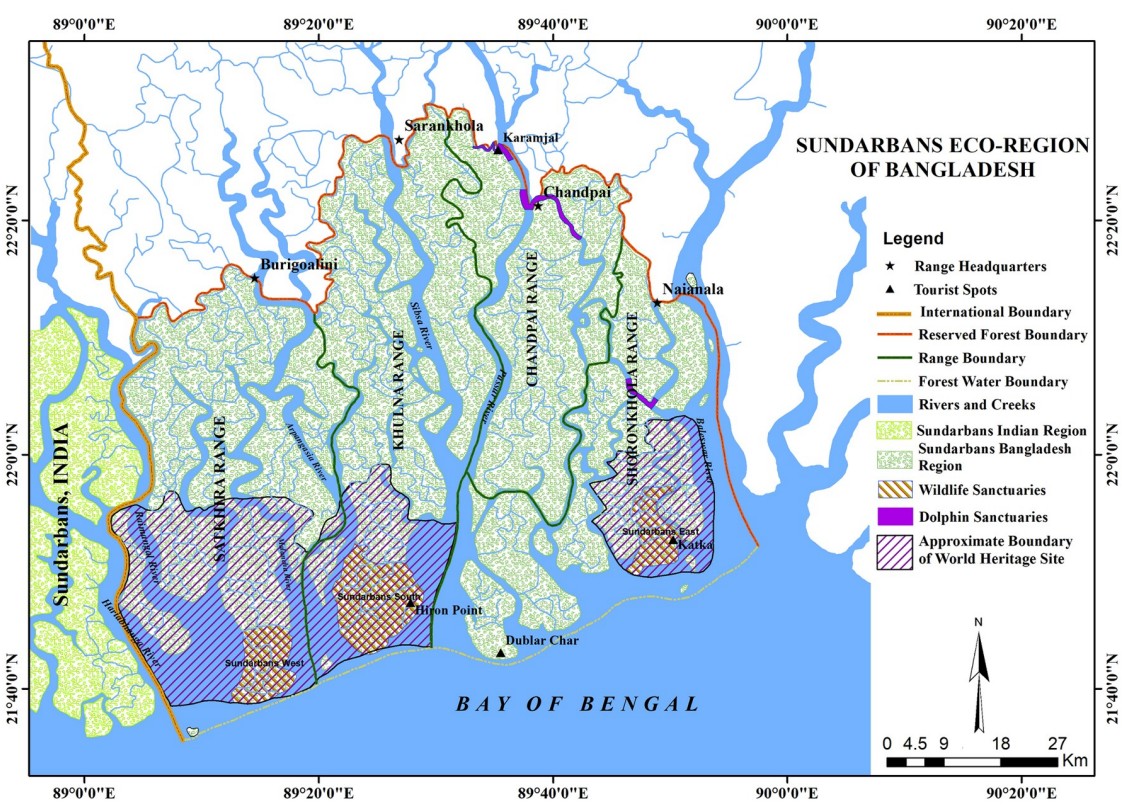

**Fig 1. Map showing the different ranges and area names of the Sundarbans where the specimens were collected.**

## Species identification

Taxonomic identification was carried out up to the species level whenever possible. Generic diagnosis, i.e., meristic counts and proportional measurements of collected specimens was accomplished following Bernacsek [40], Devi and Rao [41], Rahman et al. [42], Talwar and Kacker [43], Siddiqui et al. [44], Dev [45], Islam [46] and Siddiqui et al. [47]. All the morphological and morphometric characters observed for each sample were recorded in the premade datasheet. Fish species nomenclature was determined following the FAO Fish Identification Sheets [48].

**Genomic DNA extraction, PCR amplification, and DNA sequencing.** Two molecular markers corresponding to partial COI and 16S rRNA genes from the mitochondrial DNA were used to determine inter-specific variability. Fragments of preserved muscle or fin tissues were used for genomic DNA extraction using TIANamp Marine Animals DNA Kit (TIANGEN) following the protocol provided by manufacturer. Polymerase chain reaction (PCR) was performed for amplification of 5′ region of COI and 16S rRNA genes in a final volume of $25\mu l$ reaction mixture containing $2.5\mu l$ of reaction buffer (Green buffer), $0.5\mu l$ of dNTP mix, $16.5\mu l$ of nuclease-free water, 1μl of forward and $1\mu l$ reverse primer, $0.5\mu l$ Taq DNA polymerase, and $3\mu l$ of template DNA. The PCR primers used for amplification of the COI and 16S genes are given in Table 1. The reaction mixtures in small (0.2 ml) reaction tubes were loaded in a Thermal cycler (2720 Thermal Cycler, Applied Biosystems) for PCR. Thermal conditions of PCR varied based on using of the particular primer sets which are given in Table 1.

After successful PCR, every sample was visualized on 1.0% agarose gel stained with non-toxic fluorescent DNA dye (EZ-Vision® In-Gel Solution, USA) in a gel documentation system (Model: Syngene InGenius[3]). The flow of UV-ray was kept on to watch the band in the

**Table 1. List of primers and thermal conditions used in PCR amplification for DNA barcoding.**

| Name of Primer set | | Direction | Target region | Target organism | Author | Thermal profile |
|---|---|---|---|---|---|---|
| FishF1 | | Forward | COI | Fish | Ward et al. [15] | 95˚C (2m), 94˚C (40s), 35 cycles 54˚C (40s), 72˚C (1m) with final extension at 72˚C for (10 min) |
| FishR1 | | Reverse | | | | |
| FishF2 | | Forward | COI | Fish | | |
| FishR2 | | Reverse | | | | |
| VF2_t1 | Primer cocktails | Forward | COI | Fish | Ivanova et al. [49] | 95˚C (3m), 95˚C (30s), 35 cycles 57˚C (40s), 72˚C (1m) with final extension at 72˚C for (10 min) |
| FR1d_t1 | | Reverse | | | | |
| FishF2-t1 | | Forward | | | | |
| FishR2-t1 | | Reverse | | | | |
| 16Sar-5′ | | Forward | 16S rRNA | Fish/mollusk/crustacean | Palumbi [50] | 94˚C (3m), 94˚C (30s), 35 cycles 52˚C (40s), 72˚C (1m) with final extension at 72˚C for (10 min) |
| 16Sbr-3′ | | Reverse | | | | |
| MAXF | | Forward | COI | Crustacean | Habib et al. [10] | 95˚C (5m), 95˚C (30s), 35 cycles 42˚C (30s), 72˚C (1m) with final extension at 72˚C for (10 min) |
| MAXR | | Reverse | | | | |
| LCO1490 | | Forward | COI | Mollusca and Crustacean | Folmer et al. [51] | 95˚C (5m), 95˚C (60s), 35 cycles 52˚C (60s), 72˚C (90s) with final extension at 72˚C for (5 min) |
| HCO2198 | | Reverse | | | | |

connected computer by using GeneSys software. PCR samples with a single and clear visible band were purified using PCR Purification Kit (TIANGEN- Universal DNA Purification Kit) for sequencing. The concentration of the purified DNA was estimated by Qubit 3.0 fluorometer (Thermo Fisher Scientific). Sequencing was conducted using PCR product of the concentration of $\geq$15 pg/μl with the PCR primers by Sanger standard method in a normal automatic sequencer (Model 3730xI DNA analyzer).

**Molecular data analysis.** The obtained good consensus sequences from Sanger sequencing were selected for analysis based on chromatogram peak clarities with the help of Chromas Lit. Sequences were matched using BLAST search engine provided by NCBI and BOLD database and reciprocal BLAST through NCBI for more confirmation. Sequences were tripped by the Geneious 9.0.5 and MEGA 7.0.26 programs combined with manual proofreading and checked errors by Expasy. Each base of the spliced sequences was ensured to be correct before submitting them to BOLD database using the project names: SUN, SAUR, CRU and MSK, and to the GenBank, corresponding to accession numbers provided in S1 Table. Both of these are public-access data repositories. After that, the sequences were aligned using ClustalW program and, the parameters including the sequence length, GC content, divergence, genetic distance, transition/transversion (ti/tv) average ratio and parsimony informative sites were calculated in MEGA 6.0 software. The distances within species and between species were calculated using the Kimura 2-parameter (K2P) model [37]. Phylogenetic trees were constructed using the neighbour-joining (NJ) method. The clade credibility in the phylogeny was tested by bootstrapping, in which 10,000 repeated sampling tests were executed to obtain the support values of the clade nodes. Sequence composition and GC% in different codon positions are measured by BOLD system analyzer version[3] and MEGA 7.0.26 version. The nucleotide diversity, number of polymorphic sites and haplotype diversity were obtained using the program ARLEQUIN [52]. The COI gene sequence substitution saturation was tested using DAMBE version 7.2.25 [53].

## Results

### Genetic divergence and phylogenetic analysis of fishes

We obtained 203 sequences of 5′ region of mtDNA COI and 16S rRNA gene that were derived from 113 fish species under 105 genera, 56 families and 15 orders. Among these sequences,

**Table 2. Genetic distance of COI gene of different taxonomic levels of fish based on the K2P distance model.**

|  | Taxa | Comparisons | Min Dist(%) | Mean Dist(%) | Max Dist(%) | SE Dist(%) |
|---|---|---|---|---|---|---|
| **Within Species** | 35 | 72 | 0 | 1.57 | 20.47 | 0.06 |
| **Within Genus** | 6 | 25 | 8.2 | 15.16 | 23.64 | 0.23 |
| **Within Family** | 14 | 301 | 0 | 17.79 | 30.82 | 0.02 |

144 sequences of 93 species belonged to COI gene marker and 59 sequences of 43 species were comprised of 16S rRNA gene. Collected samples were first identified by examining morphometric characteristics and then assessed by DNA barcoding. Successfully sequenced fish species were represented by one, two or more individuals. The average length of sequence fragment of the analyzed COI barcodes was about 550 bp with no insertion, deletion or stop codon, denoting that they represented functional mitochondrial COI sequences. The COI sequences along with taxonomic files of fishes can be retrieved in the BOLD's public data portal searching by the project name SUN and NCBI GenBank accessible for all (S1 Table).

In phylogenetic analysis, the COI sequences of the same species clustered in monophyletic unit with very high bootstrap values 70–100% (S1 Fig). The sequence analysis revealed that the average nucleotide compositions in 144 COI gene sequences were A = 23.83±0.14%, T = 29.43 ±0.12%, C = 28.44±0.17% and G = 18.31±0.09%. The overall GC content was 46.74±0.2%, comparatively lower than AT content. The GC contents at the first, second and third codon positions for the 93 species of fishes were 55.78±0.13%, 42.79±0.06% and 41.66±0.55%, respectively. The overall mean distance of the sequences was 25.27±0.01%. Summary of genetic distances of different taxonomic levels viz., within species, within genera, and families based on the Kimura two-parameter (K2P) distance model is given in Table 2.

Nucleotide diversity of the entire dataset was found to be 0.211 and a total of 275 polymorphic sites were identified. The zero, two and four-fold degenerate sites were found to be 325, 41 and 70, respectively in all sequences. A total number of 133 haplotypes with diversity of 0.998 were also identified in complete dataset.

Sequence divergences for 144 COI gene sequences were compared at the species and genus levels shows in Fig 2. The distribution within-species is normalized to reduce bias in sampling at the species level, while the normalized divergence means within-species distance (%) were 1.32±0.1 (Fig 3). The proportion of transitions (S) and transversions (V) were plotted against the percent sequence divergence with the F84 genetic distance algorithm in DAMBE. The estimated ti/tv average ratio was found 1.85, which means the COI sequences generated under the

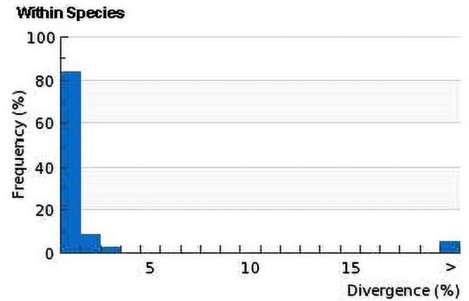 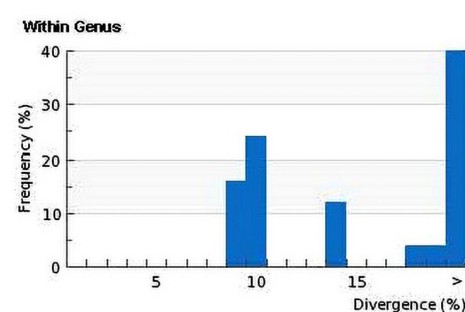

**Fig 2. Sequence divergence between all sequences at species and genus level.** Distance model: Kimura 2-Parameter, COI marker, complete deletion, GPS alignment Kalign.

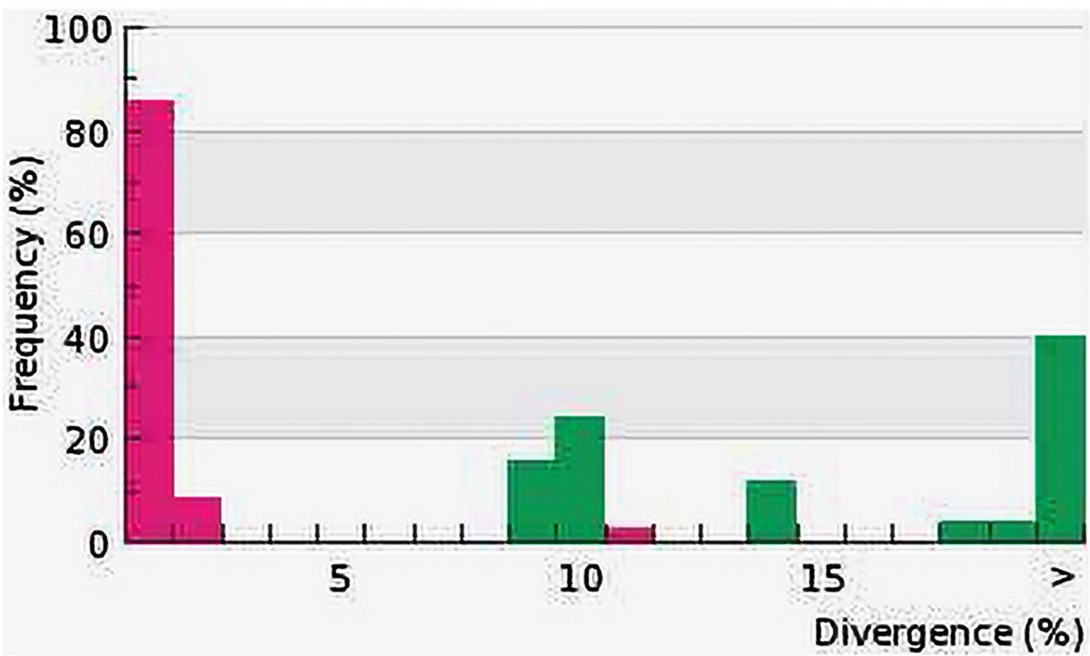

**Fig 3. Normalized divergence histogram plots shows species (pink) against the genus divergences (green).** Distance model: Kimura 2-Parameter, COI marker, Complete Deletion deletion, GPS alignment Kalign.

K2P model of 144 individuals' sequences of 93 species were not saturated and can discriminate species. Ratio of the number of transitions to the number of transversions for a pair of sequences becomes 0.5 when there is no bias towards either transitional or transversional substitution because, when the two kinds of substitution are equally probable, there are twice as many possible transversions as transitions [37]. The scatterplot (Fig 4) of COI sequences in sequence substitution saturation analysis shows that the sequences were not saturated as the transitions and transversions were linear when plotted against genetic distance (K84 distance).

Further, we sequenced 16S rRNA as a secondary gene marker for fishes that were not successfully barcoded with COI marker. Besides, in case of confusing species, we used 16S rRNA additional with COI marker. Thus, 59 individuals of 43 species belonging to 30 families of fishes were examined using 16S rRNA gene sequences. Sequence alignment of 16S rRNA gene after trimming of primer ends yielded about 500 nucleotide base pairs per taxon. The average genetic distance (%) among all sequenced individuals through 16S rRNA was estimated 14.87 ±0.01. The mean nucleotide base compositions were calculated as T = 22.33±.25%, C = 25.31 ±.29%, G = 22.16±.2%, and A = 30.2±.26%. The average genetic divergence within species, genus, and families were 1.74±.8%, 0.97±.8%, and 4.29±1.3%, respectively. The mean GC content among 59 sequences was 47.47±.38% whereas mean GC percentages at the first, second and third codon positions for the 43 species of fishes were 43.89±0.41%, 50.66±0.55% and 47.88±0.37%, respectively. In cluster analysis, the neighbor joining tree under K2P model of 16S rRNA genes for fishes clearly discriminated all the species and clustered the similar species under same nodes with significant bootstrap values 70–100% (S2 Fig). The estimated ti/tv average ratio was 1.81. Substitution pattern and rates were estimated under the K2P model (Fig 5). 16S rRNA sequences along with taxonomic files of fishes can be retrieved in the BOLD's public data portal searching by the project name SAUR.

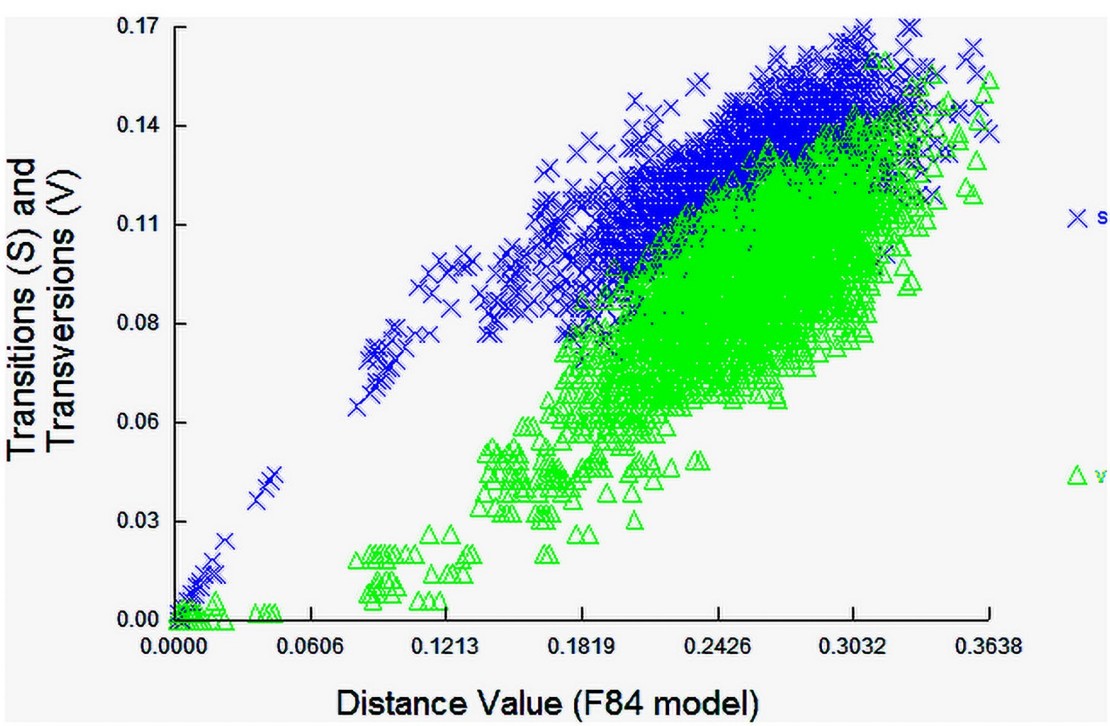

**Fig 4. Substitution saturation plot of COI gene region of fishes.** The x-axes 'F84 distance' is based on the F84 substitution model and is expected to increase linearly with divergence time. The vertical axis is for the observed proportion of transitions (S) and transversions (V), respectively.

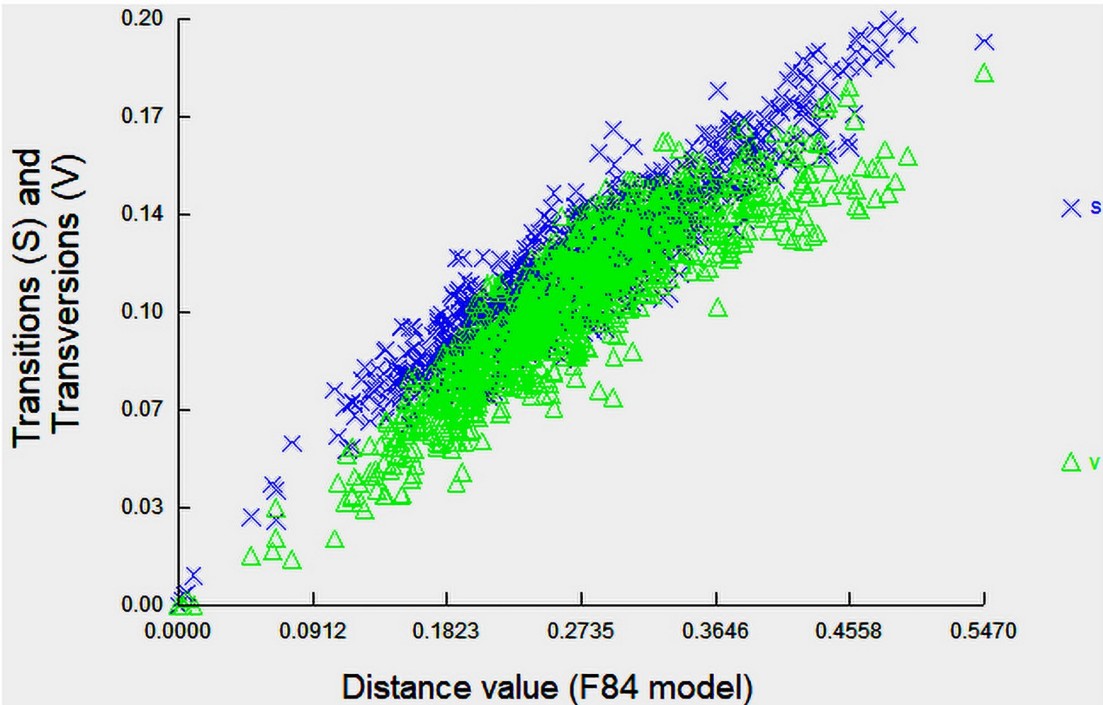

**Fig 5. Substitution saturation plot of 16S rRNA gene region of fishes.** The x-axes 'F84 distance' is based on the F80 substitution model and is expected to increase linearly with divergence time. The vertical axis is for the observed proportion of transitions (S) and transversions (V), respectively.

## Genetic divergence and phylogenetic analysis of crustaceans

Fragment lengths of the analyzed DNA barcodes (COI gene) of crustaceans ranged from 500 of 602 bp. A total of 35 crustacean individuals were collected from the Sundarbans, belonging to 13 unmistakably morphologically and genetically identified species (Table 2).

Among 13 species, 6 species belong to the crabs and rest 7 species were shrimps. These identified barcoded species belong to ten genera e.g., *Scylla*, *Portunus*, *Charybdis*, *Matuta*, *Tubuca*, *Pineaus*, *Metapenaeus*, *Mierspenaeopsis*, *Palaemon* and *Macrobrachium* of five families Portunidae, Matutidae, Ocypodidae, Penaeidae and Palaemonidae. Among all successfully sequences, *Scylla olivacea* (mud crab) represented about 31% of the COI barcodes. The mean nucleotide compositions among the barcode sequences of 35 crustacean individuals were estimated as T = 34.5±0.43%, C = 22.2±0.34%, G = 17.93±0.17%, and A = 25.46±0.25%. Mean GC content was estimated as 40.24±.47%. GC percentages at first, second, and third codon position among the barcoded sequences were 53.09±0.17%, 42.25±0.12% and 25.32±1.12%, respectively. The NJ tree revealed distinct clades separated based on the genus. The clades showed high bootstrap values ranging from 70–100% (S3 Fig). Most of these sequences were matched to reference sequences on GenBank or BOLD databases with more than 99% identity. The overall mean distance of the crustacean sequences was 21.6%. The intraspecific K2P divergences ranged from zero to 10.3% whereas interspecific distances were between 11.5% and 33.92%. As expected, a hierarchical increase in the mean Kimura 2-Parameter (K2P) genetic divergence with increasing taxonomic levels from within a species (1.4±0.03%), to within genus (17.73±0.15%), to within families (22.81±0.02%). The estimated ti/tv average ratio was 1.45 (Fig 6). The analysis involved 35 nucleotide sequences at different codon positions included first, second, third, and noncoding. Nucleotide diversity of the entire COI barcode

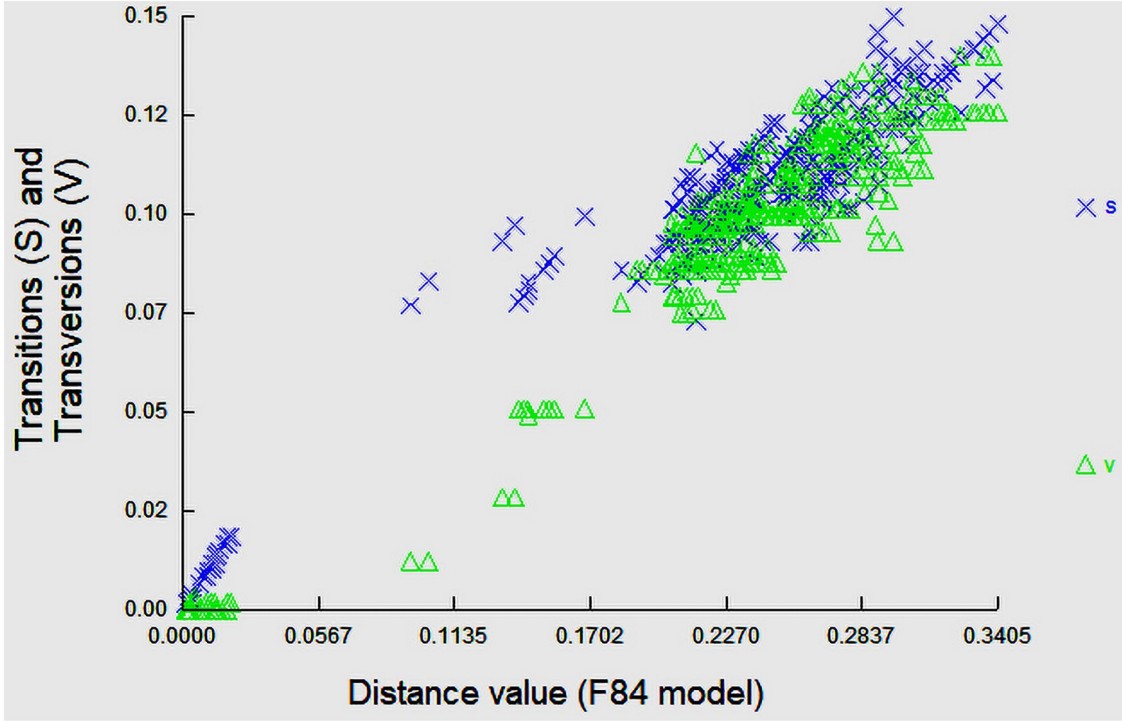

**Fig 6. Substitution saturation plot of crustacean COI sequences.** The x-axes 'F84 distance' is based on the F80 substitution model and is expected to increase linearly with divergence time.

sequence data was found to be 0.235 and a total of 326 polymorphic sites were identified. The zero, two and four-fold degenerate sites were found to be 355, 54 and 84, respectively in the sequences. A total number of 33 haplotypes with diversity of 0.997 were also identified in complete dataset. The COI sequences along with taxonomic files of crustaceans can be retrieved in the BOLD's public data portal searching by the project name CRU.

## Genetic divergence and phylogenetic analysis of molluscs

We used a total of 62 individuals for DNA barcoding represent of two specimens from each species. Unfortunately, success rate of DNA barcoding for marine and brackish water molluscs was only 10%, which represented a low number of species among morphologically confirmed species. However, only six (06) COI sequences of five species were recovered from the 62 collected specimens (S4 Fig). The average length of sequence fragment of the analyzed COI barcodes was about 500 bp with no insertion, deletion or stop codon. Values of interspecific divergence ranged from 17.43% to 66.3%. The mean nucleotide composition of the six barcoded sequences was 37.5±1.75% for T, 19.2±1.47% for C, 20.72±0.76% for G, and 22.6±0.47% for A. The mean GC content estimated as 39.9±1.8%. GC percentages at first, second, and third codon position among the barcoded sequences were 49.64±2.02%, 42.19±0.31% and 27.88 ±5.12%, respectively. Nucleotide diversity of the whole sequence data was found to be 0.283 and a total of 261 polymorphic sites were identified. The zero, two and four-fold degenerate sites were found to be 262, 25 and 11, respectively in the sequences. A total number of 5 haplotypes with diversity of 0.933 were also identified in complete dataset. The COI sequences along with taxonomic files of molluscs can be retrieved in the BOLD's public data portal searching by the project name MSK.

## Discussion

DNA barcoding has been familiarized to provide an efficient technique for species-level identifications based on the genetic distance [54]. The Sundarbans of Bangladesh provides a substantial example of on-going ecological processes as it represents the process of delta formation and the subsequent colonization of the newly formed deltaic islands and associated mangrove communities [11]. This typical geological process makes favorable habitat for many native and marine fish, as well as other aquatic lives e.g., shellfish. In this present study, we sequenced DNA barcode region of COI gene or conserved partial sequence of 16S rRNA gene for accurate identification along with morphological confirmation and building a reference library of DNA barcodes of the aquatic fauna of the Sundarbans of Bangladesh. In the present study, we sequenced 500 to 550 bp of 5′ region for COI and 16S genes for most of the species, which is well enough to use as the DNA barcodes. 5′ end of 100–150 bp COI DNA long barcode sequences (called 'DNA mini-barcode') can even efficiently identify a wide range of taxa [55]. The present study has delivered barcode coverage for 113 species of fishes. To determine the genetic distances, mtDNA CO1 and 16S rRNA genes have been used to assess inter-specific variability of different closely related congeneric fish species. Species within genera invariably clustered, and generally so did genera within families. Among the sequences, some fish species of *Atropus atropos*, *Polynemus paradiseus*, *Rachycentron canadum*, *Sciades sona*, *Thryssa hamiltonii*, *Tenualosa ilisha*, *Lagocephalus guentheri*, *Brevitrygon walga* appeared to form multiple branches under single node.

The effectiveness of species identification through DNA barcoding depends on both intraspecific divergence and interspecific divergence. DNA barcode analysis delineate the boundaries to identify species, which is related to the divergence between the closest neighbors within a group [56,57]. Generally, intraspecific divergences are rarely greater than 2% and

most are less than 1% [58]. However, there is no universal standard level for intergeneric or interfamily demarcation. In the present study, the average K2P distances of COI barcode sequences within species, genera, and families of fishes are 1.57±0.06%, 15.16±0.23%, and 17.79±0.02%, respectively. In crustaceans, the K2P distances within genera, families, and orders are 1.4±0.03%, 17.73±0.15%, and 22.81±0.02%. The fish species showed higher range of interspecific divergence from 0.19% to 36.28% whereas this range was relatively lower between 11.5% and 33.92% in case of crustaceans.

Transversions are more likely to alter the amino acid sequence of proteins than transitions, and local deviations in the ti/tv ratio are indicative of evolutionary selection on genes [59]. In our study, the estimated ti/tv average ratio was found 1.85 for COI barcode gene, and 1.81 for 16S rRNA region in fishes. On the other hand, the estimated R was 1.45 for COI barcode gene in crustacean. Variations in GC content affect different codon positions to a greater or lesser degree. The GC content among the 144 COI sequences of 93 species of fishes was higher than 35 COI sequences of the 13 species of crustaceans (46.74% versus 40.24%), largely due to GC content at the first codon position (55.78% versus 53.09%). The GC content in COI sequences at the first codon position in both taxa was the highest among three codons, which can be attributed to base usage bias among the three codon positions [60]. These differences between the codon positions indicate differences in the degree of selective constraint.

Genetically species identification is based on the principle of DNA polymorphisms, or genetic variations that take place as a result of naturally occurring mutations in the genetic code [61]. The mitochondrial COI gene has been reported as the universal DNA barcode to demarcate fish species [62]. This marker can differentiate species based on the ratio of inter- to intraspecific sequence variation [63]. On basis of genetic demarcation along with morphological confirmation, a new pufferfish species *Chelonodontops bengalensis* (Pisces, Tetraodontidae) has been described as new to science collected from the Sundarbans during the survey of present study [11]. The gene of *C. bengalensis* showed a clear separation (monophyletic clade) with other congeneric species of tetraodontids with high bootstrap support. The fish species collected from the Sundarbans, Bangladesh and barcoded in this present study were found in different category of global conservation status according to IUCN Red list of threatened species [64]. The species, namely *Ailia coila*, *Anguilla bengalensis*, *Gymnura poecilura*, and *Brevitrygon walga* collected in present study, are included under IUCN Red list global category of Near Threatened (NT) [64]. Three species of fish viz. *Megalops cyprinoides*, *Mustelus mosis*, *Scomberomorus guttatus* are incorporated under the category of Data Deficient (DD). Majority of the species which are collected and barcoded were under the category of Least Concern (LC), and Not Evaluated (NE).

The crustacean species such as shrimps, prawns, crabs, crayfish and lobsters are economically and ecologically very important groups in the Sundarbans with reporting of 28 species of prawns as the inhabitants in the Sundarbans of Bangladesh [65]. Further, 18 species of prawns and 20 species of crabs from the Sundarbans of Bangladesh were also reported [10]. However, the crustaceans seem to be more abundant in the Sundarbans of Indian part as 240 species of this group were reported from this area [66]. Beside shrimp, mud crab is a commercially important crustacean species of mangrove ecosystem in Bangladesh due to high demand and price in the international market. However, the major species of mud crab found in the Sundarbans has long been incorrectly reported as *Scylla serrata* in most of the scientific literatures in Bangladesh, even now [67–69]. COI phylogeny showed that all mud crab samples with different life stages collected in this study represented four different branches containing highly similar sequences. *S. olivacea* occurred as the most abundant species in the Sundarbans. The intraspecific K2P divergences in *Scylla olivacea* ranged from zero to 2% and the overall mean distance of the sequences was 1%. The survey of present study, genetically so far first

confirmed that the major species of mud crab of Sundarbans is *S. olivacea*, not *S. serrata* using DNA barcode analysis by COI gene [70].

Habib et al. [10] enlisted a total of 50 mollusk species in Sundarbans through morphological confirmation. Unfortunately, we were unable to successfully amplify and get good sequences for most of the molluscs samples collected in the present study. Designing new COI barcode primers (universal or family specific) may improve the success rate of PCR amplification and sequencing.

The Sundarbans is the world largest and continuous mangrove forest and supports a variety of living organisms in its surround periphery including a large number of aquatic flora and faunal biodiversity. Although the current DNA barcoding based inventory was a rapid and intensive effort to collect fish samples, and to barcode as many species as possible from the Sundarbans, this got a shortage of time to validate and confirm all of the species profiles morphologically and genetically. Some taxa are still deficit in species-level determination due to lack of data, but these will be resolved over time. We have published a pictorial identification book namely "Aquatic Biodiversity of Sundarbans, Bangladesh" where we accumulated the voucher information, taxonomic classifications, photographs, observed morphological characters, the habitats, global IUCN status, and human usage of each of the organisms collected and/or observed during the survey of present study [11]. We think this important output of the present study will perform as a valuable reference documentation on the aquatic biodiversity of Sundarbans. This is also publicly accessible as an eBook in the web address: http://www. abrlab.org/aquatic-biodiversity-of-sundarbans-bangladesh/. As an example, a picture of the page-179 containing detailed information for fish specimen *Scatophagus argus* is shown in S5 Fig.

This study has strongly authenticated the efficiency of COI and/or 16S rRNA in identifying of fish, crustaceans and molluscs species with designated barcodes. The results of this study suggest that mangrove fish fauna and crustaceans have high success rates for COI and 16S rRNA fragments amplification and single sequencing. In terms of crustacean and molluscs species, DNA amplification and single sequencing of COI and 16S rRNA fragments were relatively unsuccessful compared to the fish. In conclusion, the results from our data are consistent with different taxonomic groups, based on integrated approaches e.g., molecular and morphological characters. Those DNA barcodes data of present study can be used as an effective tool in future for accurate identification of fish eggs, larvae, and other ichthyoplanktons which are mostly and usually cryptic. Thus, it will help to detect nursery and breeding sites (e.g., canals, creeks or rivers) of important fish species in Sundarbans. The sequence data also can be used for rapid detection of fins and sliced body parts of the protected shark, skate, and ray species by Wildlife (Conservation and Security) Act 2012 to prevent illegal trading often occurred in Sundarbans [71]. This barcode library can be further expanded including other mangrove aquatic faunal and floral species to build a complete and high-coverage of the DNA barcoding database of Sundarbans. Hopefully it will contribute in future study, effective management and conservation of natural aquatic resources of the Sundarbans, Bangladesh.

## Supporting information

**S1 Fig. Neighbor-joining tree constructed using the Kimura 2-parameter model for COI gene sequences with all sequenced individual of fishes.** Bootstrap support of ≥70% are shown above branches. Scale represents genetic distance between species.
(TIF)

**S2 Fig. Neighbor-joining tree constructed using the Kimura 2-parameter model for 16S rRNA gene sequences with all sequenced individuals of fishes.** Bootstrap support of >70%

are shown above branches. Scale represents genetic distance between species. Bootstrap replications 1000.
(TIF)

**S3 Fig. Neighbor-joining tree constructed using the Kimura 2-parameter model for COI gene sequences with all sequenced crustacean individuals.** Bootstrap support of >70% are shown above branches. Scale represents genetic distance between species. Bootstrap replications 10000.
(TIF)

**S4 Fig. Neighbor-joining tree constructed using the Kimura 2-parameter model for COI gene sequences with all sequenced mollusc individuals.** Bootstrap support of >70% are shown above branches. Scale represents genetic distance between species. Bootstrap replications 10000.
(TIF)

**S5 Fig. Specimen data for the fish sample *Scatophagus argus* from the eBook "Aquatic Biodiversity of Sundarbans, Bangladesh" ([http://www.abrlab.org/aquatic-biodiversity-of-sundarbans-bangladesh/](http://www.abrlab.org/aquatic-biodiversity-of-sundarbans-bangladesh/)).**
(TIF)

**S1 Table. GeneBank accession numbers of the COI and 16S rRNA sequences used in the present study.**
(PDF)

**S1 File.**
(ZIP)

## Acknowledgments

Authors are thankful to Bangladesh Forest Department under the Ministry of Environment, Forest and Climate Change of the government of the People's Republic of Bangladesh for their strategic support during the field survey. Authors are also grateful to Korea Institute of Ocean Science and Technology for their technical support. We are grateful to Stiven Roytman, Research assistant, Kenney lab, Wayne state university, Michigan, USA for language usage, spelling, and grammar checking.

## Author Contributions

**Conceptualization:** Choong-Gon Kim.

**Data curation:** Amit Kumer Neogi.

**Formal analysis:** Amit Kumer Neogi.

**Funding acquisition:** Kazi Ahsan Habib, Youn-Ho Lee, Choong-Gon Kim.

**Investigation:** Kazi Ahsan Habib, Amit Kumer Neogi, Jina Oh.

**Methodology:** Kazi Ahsan Habib, Amit Kumer Neogi.

**Project administration:** Kazi Ahsan Habib, Youn-Ho Lee, Choong-Gon Kim.

**Resources:** Choong-Gon Kim.

**Software:** Jina Oh.

**Supervision:** Kazi Ahsan Habib.

**Validation:** Amit Kumer Neogi.

**Visualization:** Jina Oh.

**Writing – original draft:** Kazi Ahsan Habib, Amit Kumer Neogi.

**Writing – review & editing:** Kazi Ahsan Habib, Amit Kumer Neogi, Muntasir Rahman.

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
