## [Decision Letter · Decision Letter 0]

20 May 2021

PONE-D-21-08538

DNA Barcoding of Brackish and Marine Water Fishes and Shellfishes of Sundarbans, the World’s Largest Mangrove Ecosystem and UNESCO Natural Heritage Site in Bangladesh

PLOS ONE

Dear Dr. Habib,

Thank you for submitting your manuscript to PLOS ONE. After careful consideration, we feel that it has merit but does not fully meet PLOS ONE’s publication criteria as it currently stands. Therefore, we invite you to submit a revised version of the manuscript that addresses the points raised during the review process.

We look forward to receiving your revised manuscript.

Kind regards,

Bi-Song Yue, Ph.D

Academic Editor

PLOS ONE

Journal Requirements:

2.We suggest you thoroughly copyedit your manuscript for language usage, spelling, and grammar. If you do not know anyone who can help you do this, you may wish to consider employing a professional scientific editing service.  

5.We note that Figure(s) 1 in your submission contain map images which may be copyrighted. All PLOS content is published under the Creative Commons Attribution License (CC BY 4.0), which means that the manuscript, images, and Supporting Information files will be freely available online, and any third party is permitted to access, download, copy, distribute, and use these materials in any way, even commercially, with proper attribution. For these reasons, we cannot publish previously copyrighted maps or satellite images created using proprietary data, such as Google software (Google Maps, Street View, and Earth). For more information, see our copyright guidelines: http://journals.plos.org/plosone/s/licenses-and-copyright.

a)   You may seek permission from the original copyright holder of Figure(s) 1 to publish the content specifically under the CC BY 4.0 license. 

Reviewers' comments:

Reviewer's Responses to Questions

**Comments to the Author**

1. Is the manuscript technically sound, and do the data support the conclusions?

Reviewer #1: Partly

Reviewer #2: Yes

2. Has the statistical analysis been performed appropriately and rigorously? 

Reviewer #1: Yes

Reviewer #2: Yes

3. Have the authors made all data underlying the findings in their manuscript fully available?

Reviewer #1: Yes

Reviewer #2: Yes

4. Is the manuscript presented in an intelligible fashion and written in standard English?

Reviewer #1: No

Reviewer #2: Yes

5. Review Comments to the Author

Reviewer #1: Samples of fish, crabs, shrimp some mollusks were CO1 barcoded and some Mitrochondrial 16s ribosomal DNA was all sequenced for some. The data for fish seem good, crabs and shrimp ok and mollusks poor. the data are valuable for knowing what fish and crustaceans are present. A lot of data are presented (GC content and transitions vs transversions for example) but why they are presented is not well explained. What was actually done is hard to figure out because there has been no filtering. For example 3 samples were theoretically analyzed for each morphological species , but then it turns out one, two 3 or more individual individuals were collected and there is no accounting for how many out of the number sampled were successfully sequenced. The crab/shrimp data show 11 Scylla olivivacea were sequenced out of the 35 total sequences. there are data for 4 species of mollusks. the mollusk data are presented and not discussed.

Overall, the paper does not stay on track. The existing introductory sections need to be shortened and the general audience need to be introduced to additional topics (GC content - Tranversions vs transitions-divergence) be explained. The authors address CO! vs 16s in the introduction, but then do not return to it in the discussion.

Having data from this region is important and the data should be presented in a clear concise fashion. I didn't suggest reject because the data are important. The paper should be totally redone.

DNA barcoding

Line comment

1-3 suggest Title: Barcoding of Fish and Shellfish of the worlds largest mangrove ecosystem

absrtract

24-aquatic fauna , be specific looks like fish, bivalvees and crabs?

44 delete wild animals

43-63 reduce text by 40%

64-78 reduce 40%

79-98 reduce by 40% is 16s CO1 an issue of debate for your groups?

99-107 move up with story on mangroves and reduce 40%

185 188 numbers do not add up

189 to 190—you said 3 of each kind earlier—need to show data on numbers of samples somewhere.

Reviewer #2: PLOS ONE

PONE-D-21-08538

DNA Barcoding of Brackish and Marine Water Fishes and Shellfishes of Sundarbans, the World’s Largest Mangrove Ecosystem and UNESCO Natural Heritage Site in Bangladesh

The authors produced an ambitious inventory of DNA barcodes of aquatic life in the Sundarbans mangrove ecosystem. They also compared the performance of COI and 16S barcodes. The data will be useful for future biodiversity studies, development of environmental DNA assays, and taxonomy studies. The paper is well-written overall. I recommend publication with no required major revisions. The following suggestions should be considered by the authors:

Throughout, use past tense when writing about the results of the study.

Throughout, please check the significant digits in reported results. I expect most results reported as percentages should have 3 significant digits, not 4.

Abstract, line 22: Change “apply DNA barcoding tool” to “apply a DNA barcoding tool”.

Abstract, line 25: Change “sequences of 16S rRNA gene” to “sequences of the 16S rRNA gene”

Abstract, line 26: The reference to “3 aquatic taxa” is not very precise. I’d suggest changing “3 aquatic taxa (fish, crustaceans and mollusk)” to “aquatic fish, crustaceans, and molluscs”

Abstract: Give the standard deviations for all the average K2P distances reported.

Abstract, lines 36-37: Change “misidentification of mud crab species of Sundarbans” to “misidentification of a mud crab species of the Sundarbans”.

Abstract, line 37: Change “In case of molluscs” to “In the case of molluscs”

Abstract, lines 38-40: Change last sentence to “The present study describes the development of a molecular and morphometric cross-referenced inventory of the fish and shellfish of the Sundarbans. This inventory will be useful in future biodiversity studies and in informing future conservation plans.

Introduction, line 56: I think instead of “exclusively adapted” you mean “uniquely adapted”.

Introduction, line 64: Change “The Sundarbans is the world’s single largest 64 continuous mangrove forest lies…” to “The Sundarbans, the world’s single largest 64 continuous mangrove forest, lies…”

Introduction, line 76: Change “All of these factors educe urgent need…” to “All of these factors highlight the urgent need…”

Introduction, line 80: Change “under the term’s DNA barcoding or DNA” to “under the terms DNA barcoding or DNA taxonomy”

Materials and Methods, lines 151-153: Please give final concentrations of PCR components rather than volumes.

Materials and Methods, line 157: Do you mean a 1.0 % agarose gel?

Materials and Methods, line 168: Was there any reciprocal BLAST or other error checking of the database matches?

Results, line 222: Delete “respective”, it is not needed here.

Results, lines 268-269: Were the mollusk samples tested for PCR inhibition? If the samples were inhibited, a second DNA extraction, or 10-fold dilution of the DNA sample might improve the sequencing results.

Discussion, line 283: Delete “among entities within species”

Discussion, line 284: Change “runs” to “provides”

6. PLOS authors have the option to publish the peer review history of their article (what does this mean?). If published, this will include your full peer review and any attached files.

Reviewer #1: No

Reviewer #2: No

---

## [Author Response · Author response to Decision Letter 0]

6 Jul 2021

Dear Reviewer,

Thank you so much for your kind feedback to improve of our manuscript. We believe that, the current changes will serve the purpose for publication. We hope and believe that the present study and generated data will help in the study of other investigators such as taxonomists, biologists and conservationists for further research in this unique coastal ecosystem especially in the South and South-east Asia.

Look forward for your kind approval to publish this article.

---

## [Editor Report · Decision Letter 1]

12 Jul 2021

DNA barcoding of brackish and marine water fishes and shellfishes of Sundarbans, the world’s largest mangrove ecosystem

PONE-D-21-08538R1

Dear Dr. Habib,

We’re pleased to inform you that your manuscript has been judged scientifically suitable for publication and will be formally accepted for publication once it meets all outstanding technical requirements.

Kind regards,

Bi-Song Yue, Ph.D

Academic Editor

PLOS ONE

---

## [Editor Report · Acceptance letter]

23 Jul 2021

PONE-D-21-08538R1 

DNA barcoding of brackish and marine water fishes and shellfishes of Sundarbans, the world’s largest mangrove ecosystem 

Dear Dr. Habib:

I'm pleased to inform you that your manuscript has been deemed suitable for publication in PLOS ONE. Congratulations! Your manuscript is now with our production department. 

Kind regards, 

on behalf of

Dr. Bi-Song Yue 

Academic Editor

PLOS ONE